# ActAnywhere: Subject-Aware Video Background Generation

**Boxiao Pan**[*]
Stanford University

**Zhan Xu**
Adobe Research

**Chun-Hao Paul Huang**
Adobe Research

**Krishna Kumar Singh**
Adobe Research

**Yang Zhou**
Adobe Research

**Leonidas J. Guibas**
Stanford University

**Jimei Yang**
Runway

## Abstract

We study a novel problem to automatically generate video background that tailors to foreground subject motion. It is an important problem for the movie industry and visual effects community, which traditionally requires tedious manual efforts to solve. To this end, we propose ActAnywhere, a video diffusion model that takes as input a sequence of foreground subject segmentation together with an image of a novel background, and generates a video of the subject interacting in this background. We train our model on a large-scale dataset of 2.4M videos of human-scene interactions. Through extensive evaluation, we show that our model produces videos with realistic foreground-background interaction while strictly following the guidance of the condition image. Our model generalizes to diverse scenarios including non-human subjects, gaming and animation clips, as well as videos with multiple moving subjects. Both quantitative and qualitative comparisons demonstrate that our model significantly outperforms existing methods, which fail to accomplish the studied task. Please visit our project webpage at `https://actanywhere.github.io`.

## 1   Introduction

Compositing an acting video of a subject onto a novel background is central for creative story-telling in filmmaking and visual effects. The key requirement is seamlessly integrating the foreground subject with the background in terms of camera motions, interactions, lighting and shadows, so that the composition looks realistic and vivid as if the subject acts physically in the scene. In movie industry, this process is often conducted by virtual production [1] that requires artists to first create a 3D scene and then to film the acting video in an LED-walled studio or to render the video in 3D engines. This process is not only tedious and expensive, but most importantly, prevents artists from quickly iterating their ideas.

Inspired by this artistic workflow, we study the novel problem of *automated* subject-aware video background generation. As shown in fig. 1, given a foreground segmentation sequence that provides the subject motion, as well as a condition frame that describes a scene, we aim to generate a video that adapts the subject to this scene with realistically synthesized foreground-background interaction. This condition frame can be either a background-only image, or a composite frame consisting of both background and foreground created via photo editing tools [3] or generative image model [32].

This problem, at its core, requires retaining part of the input video while generating the rest to adapt to it. To the best of our knowledge, the closest works to this setting are those on video editing and inpainting / outpainting. Video editing methods assume a source video as input and make edits

---

[*]Work done during an internship at Adobe.

38th Conference on Neural Information Processing Systems (NeurIPS 2024).

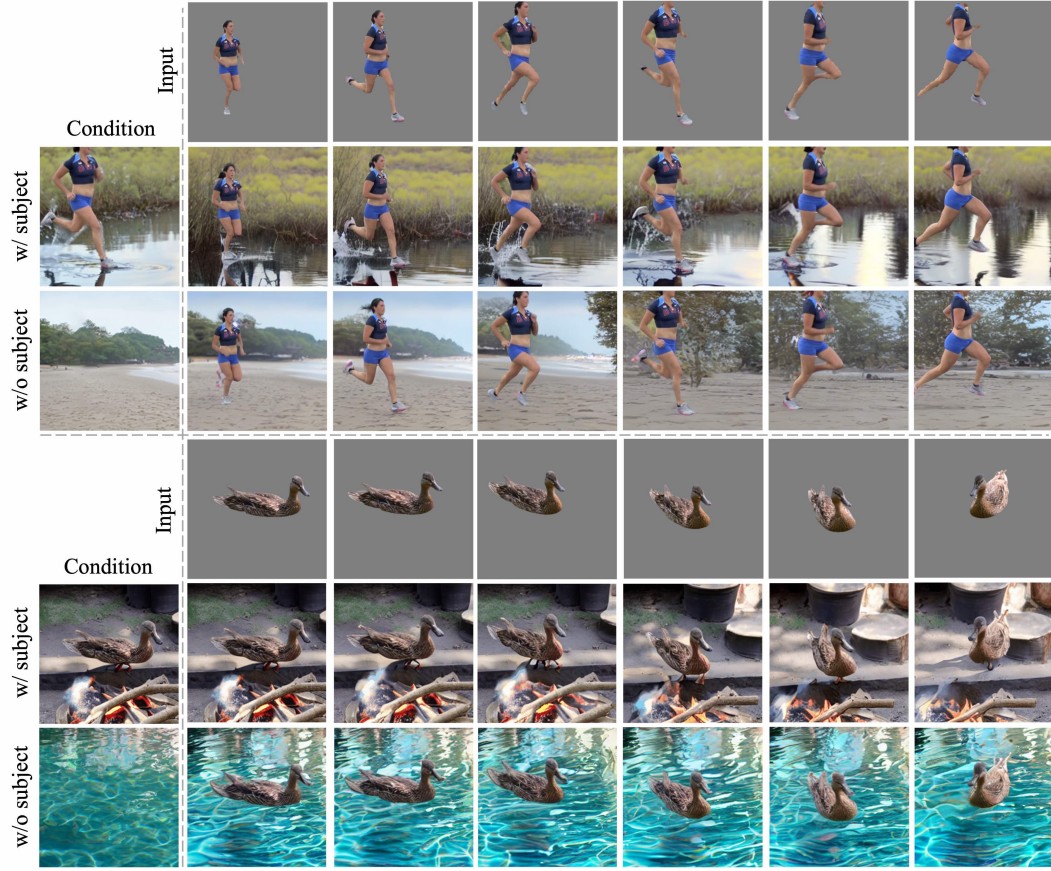

Figure 1: Given a sequence of foreground segmentation as input (top), and one frame that describes the background as the condition (left), ActAnywhere generates coherent video background that adapts to the subject motion. We show two subjects here, each with two generated samples. ActAnywhere is able to generate videos consistent with the condition frame with highly realistic details such as splatting water, moving smoke and flame, shadows, duck feet, etc. It generalizes to a diverse distribution of subjects and backgrounds, including non-human subjects. Our method works with both composited frames and background-only images as the condition.

based on condition signals, such as natural language or image. In comparison, our task poses unique challenges that these methods cannot solve. First, the foreground of the video needs to be retained and sometimes refined on the boundary (as shown in fig. 8 in appendix), while prior works generally perform holistic changes to the entire video [5, 8, 13, 16, 22, 24, 42]. Second, existing methods most commonly condition the editing process on a short text prompt [5, 8, 13, 16, 22, 24, 42], which only poses loose constraints that are not enough for artists' creative intentions. Those that can condition on an image [13] are not able to generate contents that strictly follow the guidance. We demonstrate such comparisons in fig. 4. On the other hand, video inpainting / outpainting methods [14, 43, 46, 49] aim to perform context-aware removal / expanding for a video. However, these methods focus on pixel harmonization, while our emphasis is on taking additional control signal that indicates the desired background, and synthesizing large and dynamic background region with reasonable interaction with the given foreground.

To this end, we propose a video diffusion model that explicitly controls the foreground motion with the foreground segmentation sequence, and additionally, conditions the generation on an image that describes the desired background. We train the model with a designed self-supervised learning procedure that takes the foreground segmentation sequence of the input video to predict the original video, conditioned on a randomly sampled frame. This training procedure enables the model to 1) retain the foreground subject; 2) hallucinate missing details from imperfect segmentation; and 3) adhere to the guidance of the condition frame while being robust to the subject's pose in it.

Moreover, we propose to express the condition signal as the CLIP features of the image, which we show empirically outperforms its alternatives. We also concatenate a background mask with the segmentation as the input to better indicate the region to be generated.

We train our model on a large-scale dataset [26] that consists of 2.4M videos of human-scene interactions and evaluate both on a held-out set as well as on videos from DAVIS [30]. ActAnywhere is able to generate highly realistic videos that follow the condition frame, and at the same time synthesizes video background that conforms to the foreground motion. Notably, despite being trained solely on videos of humans, ActAnywhere generalizes to non-human subjects, such as animals and man-made objects, in a zero-shot manner.

In summary, our contributions are:

1. We introduce a novel problem of automated subject-aware video background generation.
2. We propose ActAnywhere, a video diffusion model to solve this task, and train it on a large-scale human-scene interaction video dataset in a self-supervised manner.
3. Extensive evaluations demonstrate that our model generates coherent videos with realistic subject-scene interactions, camera motions, lighting and shadows, and generalizes to out-of-distribution data including non-human subjects, gaming and animation clips, and videos with multiple moving subjects.

## 2  Related Work

**Video generation**.  There have been a long thread of works on video generation. The core architecture has evolved from GANs [11, 38, 41] to more recent transformers [15, 40, 46, 49] and diffusion models [6, 9, 17, 19, 20, 24, 47]. Below we review the most related diffusion-based works. Most of these works leverage temporal self-attention blocks inside the denoising U-Net in order to acquire temporal awareness. On top of that, Text2Video-Zero [24] introduces additional noise scheduling to correlate the latents in a video. LVDM [19] and Align Your Latents [6] both design a hierarchical approach to generate longer-term videos. Align Your Latents additionally fine-tunes a spatial super-resolution model for high-resolution video generation. AnimateDiff [17] proposes to train the temporal attention blocks on a large-scale video dataset, which can then be inserted into any text-to-image diffusion models (given that the architecture fits) to turn that into a text-to-video model, in a zero-shot manner. VideoCrafter1 [9] further uses dual attention to enable joint text and image-conditioned generation. These works focus on unconditional generation or with text or image conditioning, but are not able to follow the guidance of additional foreground motion.

**Video editing**. Another thread studies the problem of video editing, where a source video is given as input, and edits are performed according to some condition signals. Text2Live [5] uses pre-trained video atlases of the input video, and performs text-guided edits on the foreground or background. Gen1 [13] leverages depth maps estimated by a pre-trained network [33] as an additional condition to improve the structural consistency. Tune-A-Video [42] proposes to finetune only part of the spatial-attention blocks and all of the temporal-attention blocks on a single input video. TokenFlow [16] uses latent nearest neighbor fields computed from the input video to propagate edited features across all frames. Both VideoControlNet [22] and Control-A-Video [10] adopt a ControlNet [48]-like approach to condition the video diffusion process with additional signals such as depth maps or Canny edges extracted from the input video. As stated above, these works apply holistic changes to the entire video and are not able to retain the subject. Moreover, they either take only loose constraints from text conditioning, or are not able to strictly follow the image guidance.

One major downside of these works is hence that the generated videos tend to keep the spatial structure from the source video, which limits the edits that the model can perform to stylistic changes. In our work, we propose to condition on the foreground segmentation for the motion, while extract the background information only from one condition frame. In particular, using the masked foreground as input endows a nice separation as in what to preserve and what to generate.

**Image and video inpainting**.  Image / video inpainting aims to fill a missing region, often expressed as a mask. Image inpainting methods either take condition signals such as natural language and image [34, 44, 45], or rely solely on the context outside the masked region [14, 36, 46, 49]. Recent diffusion-based image inpainting methods use a combination of masked image and the mask itself, and condition the diffusion process either on natural language [34, 44] or an image of the condition

object [45], or perform unconditional diffusion [36]. For video inpainting, MAGVIT [46] proposes a generative video transformer trained through masked token prediction, and is able to inpaint small masked regions afterwards. ProPainter [49] designs a flow-based method by propagating pixels and features through completed flows. M3DDM [14] leverages a video diffusion model, and conditions the diffusion process on global video features extracted by a video encoder. Different from these works, we aim to generate large background regions that strictly follow the condition frame. Moreover, the generated background needs to adapt to the foreground subject motion in a coherent way. This poses significant challenges that previous inpainting methods cannot tackle.

## 3 Method

We first provide essential preliminary background on latent diffusion in section 3.1. We then formally define our problem in section 3.2 and delve into our model design in section 3.3. Finally, we specify the training details in section 3.4.

### 3.1 Preliminaries on Latent Diffusion Models

Diffusion models such as DDPM [21], encapsulate a forward process of adding noise and a backward process of denoising. Given a diffusion time step $\tau$, the forward process incrementally introduces Gaussian noises into the data distribution $x_0 \sim q(x_0)$ via a Markov chain, following a predefined variance schedule denoted as $\beta$:

$$q(\mathbf{x}_\tau | \mathbf{x}_{\tau-1}) = \mathcal{N}(\mathbf{x}_\tau; \sqrt{1 - \beta_\tau} \mathbf{x}_{\tau-1}, \beta_\tau \mathcal{I}) \tag{1}$$

For the backward process, a U-Net [35] $\epsilon_\theta$ is trained to denoise $\mathbf{x}_\tau$ and recover the original data distribution:

$$p_\theta(\mathbf{x}_{\tau-1} | \mathbf{x}_\tau) = \mathcal{N}(\mathbf{x}_{\tau-1}; \boldsymbol{\mu}_\theta(\mathbf{x}_\tau, \tau), \boldsymbol{\Sigma}_\theta(\mathbf{x}_\tau, \tau)) \tag{2}$$

$\boldsymbol{\mu}_\theta$ and $\boldsymbol{\Sigma}_\theta$ are parametrized by $\epsilon_\theta$. The discrepancy between the predicted noise and the ground-truth noise is minimized as the training objective.

Stable Diffusion [34] further proposes to train the diffusion model in the latent space of a VAE [25]. Specifically, an encoder $\mathcal{E}$ learns to compress an input image $x$ into latent representations $z = \mathcal{E}(x)$, and a decoder $\mathcal{D}$ learns to reconstruct the latents back to pixel space, such that $x = \mathcal{D}(\mathcal{E}(x))$. In this way, the diffusion is performed in the latent space of the VAE.

### 3.2 Problem Formulation

Given an input video $\mathcal{X} \in \mathbb{R}^{T \times H \times W \times 3}$ featuring a foreground subject, we first deploy a segmentation algorithm, such as Mask R-CNN [18], to obtain a subject segmentation sequence, $\mathcal{S} \in \mathbb{R}^{T \times H \times W \times 3}$, along with the corresponding masks, $\mathcal{M} \in \mathbb{R}^{T \times H \times W \times 1}$. Both $\mathcal{S}$ and $\mathcal{M}$ serve as input to our model. $\mathcal{S}$ contains the segmentation of the foreground subject, with background pixels set to 127 (grey). $\mathcal{M}$ has the foreground pixels set to 0 and background to 1. Across all our experiments, $H = W = 256$ and $T = 16$.

Additionally, we also incorporate a single condition frame $\mathbf{c} \in \mathbb{R}^{H \times W \times 3}$ describing the background that we want to generate. As shown in fig. 2, $\mathbf{c}$ is a randomly sampled frame from $\mathcal{X}$ at training time, while can be either a frame showing foreground-background composition or a background-only image at inference time. The goal is thus to generate an output video $\mathcal{V}$ with the subject dynamically interacting with the synthesized background. The motivation of using an image not language as the condition is that image is a more straightforward media to carry detailed and specific information of the intended background, especially when users already have a pre-defined target scene image.

### 3.3 Subject-Aware Latent Video Diffusion

We build our model based on latent video diffusion models [17]. In our architecture design, we address two main questions: 1) providing the foreground subject sequence to the network to enable proper motion guidance, and 2) injecting the condition signal from the background frame to make the generated video adhere to the condition.

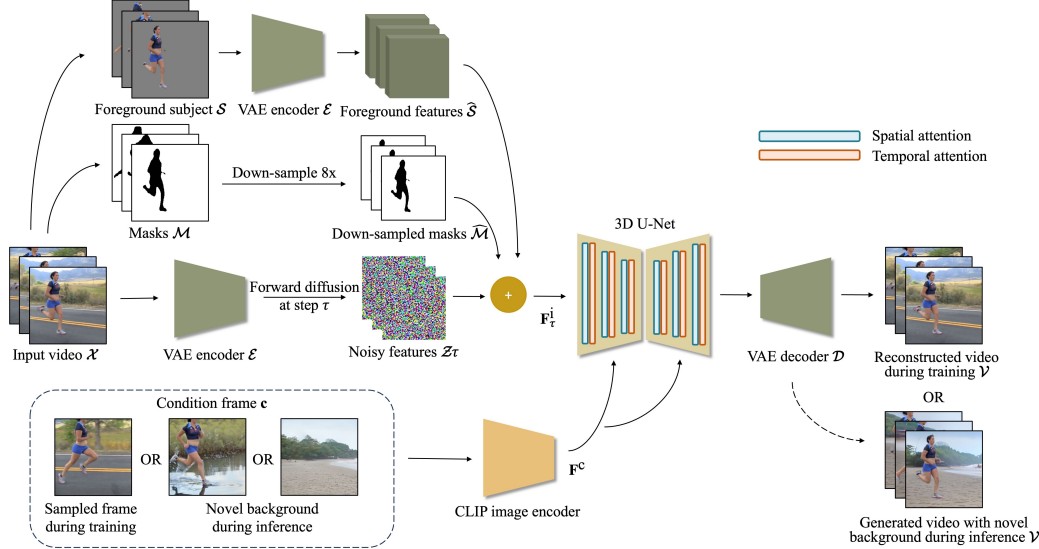

Figure 2: **Architecture overview**. During training, we take a randomly sampled frame from the training video to condition the denoising process. At test time, the condition can be either a composited frame of the subject with a novel background, or a background-only image.

We present our pipeline in fig. 2. For the foreground segmentation sequence $\mathcal{S}$, we use the pre-trained VAE [34] encoder $\mathcal{E}$ to encode the foreground segmentation into latent features $\hat{\mathcal{S}} \in \mathbb{R}^{16 \times 32 \times 32 \times 4}$. We downsample the foreground mask sequence $\mathcal{M}$ 8 times to obtain the resized mask sequence $\hat{\mathcal{M}} \in \mathbb{R}^{16 \times 32 \times 32 \times 1}$ to align with the latent features $\hat{\mathcal{S}}$. To train the denoising network $\epsilon_\theta$, we encode the original frames $\mathcal{X}$ with the same VAE encoder into latent representation $\mathcal{Z} \in \mathbb{R}^{16 \times 32 \times 32 \times 4}$, and add noises at diffusion time step $\tau$ with the forward diffusion process denoted in eq. (1) to get noisy latent feature $\mathcal{Z}_\tau$. We subsequently concatenate $\hat{\mathcal{S}}$, $\hat{\mathcal{M}}$ and $\mathcal{Z}_\tau$ along the feature dimension, forming a 9-channel input feature $\mathbf{F}_\tau^i \in \mathbb{R}^{16 \times 9 \times 32 \times 32}$ to the U-Net. During inference, $\mathcal{Z}_0$ is initialized as Gaussian noises, and gets auto-regressively denoised for multiple time steps to sample a final result, according to the backward diffusion process described in eq. (2). The denoised latents are then decoded to a video via the VAE decoder $\mathcal{D}$.

We build our 3D denoising U-Net based on AnimateDiff [17]. AnimateDiff works by inserting a series of motion modules in between the spatial attention layers in the denoising U-Net of a pre-trained T2I diffusion model. These motion modules consist of a few feature projection layers followed by 1D temporal self-attention blocks.

For the condition image $\mathbf{c}$, we encode it with the CLIP image encoder [31] and take the features from the last hidden layer as its encoding $\mathbf{F}^c$. These features are then injected into the UNet $\epsilon_\theta$ through its cross-attention layers, similar to [26, 34]. We empirically find that this method achieves better temporal consistency compared to other alternatives, such as using VAE features for either cross-attention or concatenation with other input features. We ablate on this in table 3.

### 3.4 Training

Training is supervised by a simplified diffusion objective, namely predicting the added noise [21]:

$$\mathcal{L} = ||\epsilon - \epsilon_\theta(\mathbf{F}_\tau^i, \tau, \mathbf{F}^c)||_2^2 \tag{3}$$

where $\epsilon$ is the ground-truth noise added.

**Dataset**. We train on the large-scale dataset compiled and processed by [26], which we refer to as HiC+. The resulting dataset contains 2.4M videos of human-scene interactions. It also provides foreground segmentation and masks. We refer the reader to the original paper for more details.

**Pre-trained weights**. We initialize the weights of our denoising network $\epsilon_\theta$ with the pre-trained weights from the Stable Diffusion image inpainting model [34][†], which is fine-tuned on top of the original Stable Diffusion on the text-conditioned image inpainting task. We initialize the weights of the inserted motion modules with AnimateDiff v2[‡].

For the CLIP image encoder, we use the "clip-vit-large-patch14" variant[§] provided by OpenAI, whose features from the last hidden layer have a dimension of 1024, while the pre-trained U-Net takes in features of dimension 768 as the condition, which are also in the text feature space. To account for this, we train an additional two-layer MLP to project the features into the desired space.

During training, we freeze the shared VAE and the CLIP encoder, and fine-tune the entire U-Net.

**Data processing and augmentation**. Obtaining perfect segmentation masks from videos is challenging. The masks may be incomplete, missing some parts of the foreground, or be excessive such that they include leaked background near the boundary. To deal with incomplete segmentation, during training, we apply random rectangular cut-outs to the foreground segmentation and masks. We provide more information on this in section 7.2 of the appendix. To reduce information leak from excessive segmentation, we perform image erosion to the segmentation and masks with a uniform kernel of size $5 \times 5$, both during training and inference.

**Random condition dropping**. In order to enable classifier-free guidance at test time, we randomly drop the segmentation and the mask, the condition frame, or all of them at 10% probability each during training. In these cases we set them to zeros before passing into the respective encoders.

**Other details**. We use the AdamW [27] optimizer with a constant learning rate of 3e-5. We train on 8 NVIDIA A100-80GB GPUs with batch size 4, which takes approximately a week to fully converge.

## 4 Experiments

We start by describing the data used for evaluation. We then show diverse samples generated from our method in section 4.1, both using an inpainted frame and a background-only frame as the condition. In section 4.2 and section 4.3, we compare with baselines through qualitative and quantitative evaluations, including a user study. In section 4.4, we provide ablation study results on key design choices. In the appendix, we include additional results on general video inpainting / outpainting, generalization to videos from diverse domains, and robustness to inaccurate segmentation, along with further implementation details and a discussion on limitations and potential ethical impacts.

We strongly encourage the reader to check our webpage, where we show extensive videos on video background generation with diverse generated contents and camera motions, and under various condition scenarios. It also contains the video version of the comparison with baselines.

**Evaluation data**. Following prior works [5, 10, 13, 16, 42], we compare with previous works on videos sampled from the DAVIS [30] dataset. We select videos with both human and non-human subjects. We also evaluate qualitatively and perform ablation study on held-out samples from the HiC+ dataset following the original data splits [26]. Samples with our method are generated with 50 denoising steps, with a guidance scale [34] of 5.

### 4.1 Diverse Generation with ActAnywhere

In fig. 3, we show results on the held-out segmentation sequences from the HiC+ dataset, using an inpainted frame or a background-only frame as condition. ActAnywhere generates highly realistic foreground-background interactions both at coarse and fine levels. At a coarse level, our model synthesizes road structure, pumpkin field, city views, waves, etc. that align with the subject's motion. While at a fine level, our method also generates small moving objects that are in close interaction with the subject, such as the buckets, bed sheets, horses and dune buggies, as well as the dog. Moreover, these generation stay consistent across frames, and tightly follow the guidance in the condition frame. The synthesized backgrounds also exhibit coherent scale, lighting, and shadows (also see fig. 1).

---

[†]https://huggingface.co/runwayml/stable-diffusion-inpainting
[‡]https://github.com/guoyww/animatediff/
[§]https://huggingface.co/openai/clip-vit-large-patch14

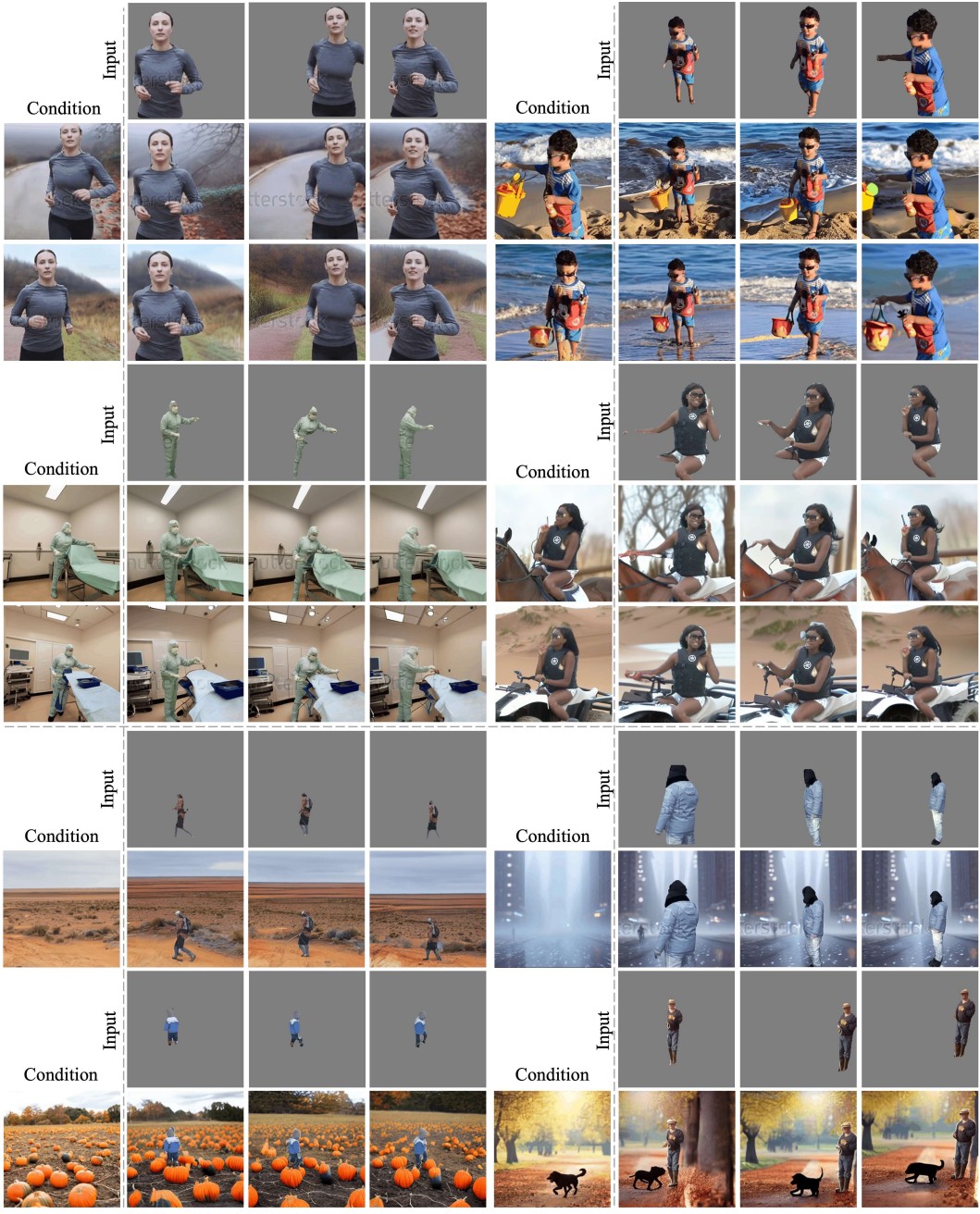

Figure 3: **Diverse results from our method**. The top part shows examples using inpainted frames as condition, while bottom contains examples with background-only conditioning. Foreground sequences are from the held-out set of HiC+.

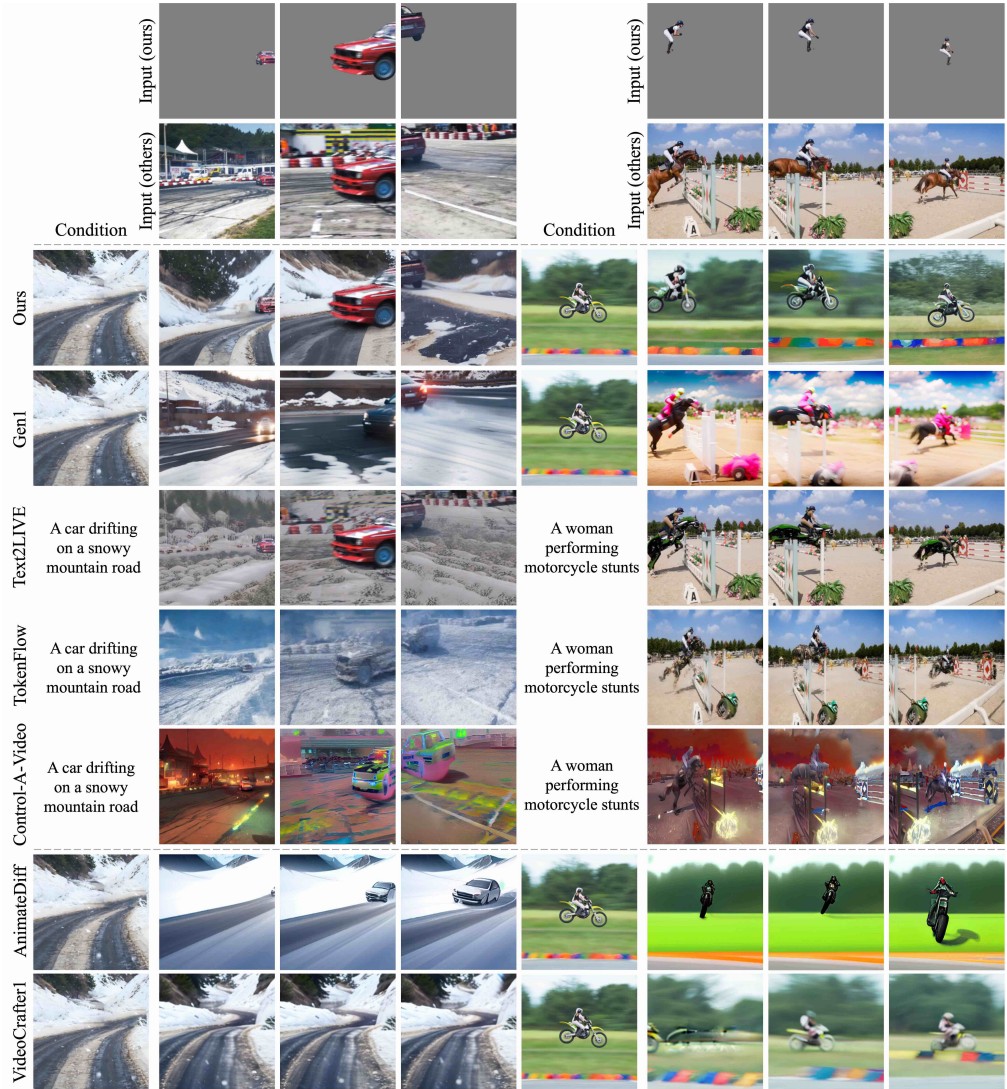

Figure 4: **Comparison with baselines.** We provide results on two videos sampled from the DAVIS [30] dataset. For each example, we show three representative frames (top) and their corresponding condition signal (left). Note that different methods assume different input, conditioning or pre-trained models, as specified in section 4.2.

## 4.2 Qualitative Comparison

**Baselines**. We first clarify that since we study a *novel problem*, and there is no prior work operating under the exact same setting to the best of our knowledge. We hence compare to closest works and adapt some, *i.e*. AnimateDiff [17], if necessary. Nonetheless, we emphasize that the formulation and pipeline are the core contribution of this work.

We compare ActAnywhere to a number of baselines, which we classify based on whether they do (fig. 4 middle) or do not (fig. 4 bottom) take a video as input. For the methods taking a video as input, Gen1 [13] takes an additional image as condition, and also leverages a pre-trained depth-estimation network [33]. Given pre-trained neural atlases [23], Text2LIVE [5] assumes a text prompt as condition to synthesize the edited video. TokenFlow [16] also uses text conditioning. Control-A-Video [10] first extracts Canny edges from the input video, then synthesizes the output video conditioned jointly on the edges and text.

| Method | CLIP_cond ↑ | CLIP_temp ↑ | FVD ↓ |
|---|---|---|---|
| Control-A-Video [7] | 0.643 | 0.942 | 381.8 |
| TokenFlow [12] | 0.762 | 0.943 | 323.0 |
| Gen1 [9] | 0.827 | 0.943 | 337.2 |
| Ours | **0.862** | **0.945** | **313.4** |

Table 1: **Quantitative comparison** against baselines.

| Condition | Visual | Temporal |
|---|---|---|
| 97.47% | 72.13% | 64.36% |

Table 2: **User study**

For baselines that do not take a video as input, the original AnimateDiff [17] only uses text conditioning. We use the strategy contributed by a public pull request[¶] to make it take additional image conditioning. Specifically, at test time, latent features are first extracted from the condition image with the pre-trained SD VAE encoder [34], which are then merged with the original per-frame Gaussian noises through linear blending. The diffusion process is later conditioned on a text prompt too. VideoCrafter1 [9] provides both a text-to-video and an image-to-video model. We use the latter for a closer comparison setting.

**Results**. The qualitative comparison on two examples from the DAVIS [30] dataset is shown in fig. 4. Our method generates temporally coherent videos that follow the foreground motion with highly realistic details, *e.g.* falling snow and snow on the car windshield, while strictly following the guidance and constraints given by the condition frame. Baseline methods in the first category generally inherit the structure present in the input video, *e.g.* road direction, horse, etc., and hence they completely fail when fine-grained edits are desired, *e.g.* horse changes to motorcycle in the second case. Methods in the second category generate unconstrained motion due to lack of guidance (VideoCrafter1 in the second example generates backward motion, which is more evident in the video on our webpage).

## 4.3 Quantitative Comparison

**Baselines**. We compare with Control-A-Video [10], TokenFlow [16] and Gen1 [13]. We also conduct a user study for preferences in comparison to Gen1.

**Metrics**. We evaluate the consistency to the condition image, the temporal consistency of the generated videos, as well as the general generation quality. Specifically, we report the average cosine similarity between the CLIP [31] image embeddings of all generated frames and that of the condition image (*CLIP_cond*), between all pairs of generated frames (*CLIP_temp*), and the FVD score [39] against a set of real videos.

For the user study, we ask the participants if they prefer results from our model vs. those from Gen1 on: 1) consistency to condition image; 2) visual quality; and 3) temporal consistency. The results are presented as the percentage of our model being preferred over Gen1.

**Results**. Results on 30 videos from DAVIS are reported in table 1. Our model outperforms baselines across all metrics, particularly by a big margin on *CLIP_cond*, suggesting that our model is able to generate videos that tightly follow the guidance of the condition frame. For the user study, we randomly select 20 videos and ask 16 participants for their preference over Gen1 [13]. Results are shown in table 2. Our method is strongly preferred on these key aspects.

## 4.4 Ablation Study

We study different choices of conditioning by experimenting with three variants. *VAE Concat* uses the same VAE encoder $\mathcal{E}$ to extract features for the condition frame $\mathbf{c}$, and concatenates with the input $\mathbf{F}_\tau^i$ along the feature dimension. *VAE Cross-Attn* leverages the VAE features $\mathcal{E}(\mathbf{c})$ through cross-attention instead of concatenation. *No Mask* does not concatenate the masks $\mathcal{M}$ with the foreground segmentation $\mathcal{S}$ and noise $\mathcal{Z}_\tau$ for the input.

The results on the held-out set of HiC+ are shown in table 3. Providing the condition in the CLIP feature space through cross-attention provides global semantic guidance to the entire diffusion process, while VAE features pose stricter spatial constraints, especially through concatenation, hence making it harder to produce coherent videos that conform to the condition. Providing the mask as

---

[¶]https://github.com/guoyww/AnimateDiff/pull/8

| Variant | CLIP_cond ↑ | CLIP_temp ↑ | FVD ↓ |
|---|---|---|---|
| VAE Concat | 0.743 | 0.939 | 335.2 |
| VAE Cross-Attn | 0.823 | 0.942 | 326.6 |
| No Mask | 0.850 | 0.943 | 321.7 |
| Full Model (Ours) | **0.857** | **0.944** | **315.2** |

Table 3: **Ablation study** on key design choices.

additional input further indicates the region that the model should generate thus also improves the overall generation performance.

## 5   Conclusion

We present ActAnywhere, a video diffusion-based model that generates videos with coherent and vivid foreground-background interactions, given an input foreground segmentation sequence and a condition frame describing the background. Our model synthesizes highly realistic details such as moving or interacting objects and shadows. The generated videos also exhibit consistent camera scales and lighting effects. We believe our work contributes a useful tool for the movie and visual effects community, as well as for the general public to realize novel ideas of situating an acting subject in diverse scenes, in a simple and efficient way that is not previously possible.

# 6 Acknowledgment

We thank the authors of [26] for compiling and processing the dataset HiC+, especially Sumith Kulal for the code and instructions on accessing the data. We also thank Jiahui (Gabriel) Huang from Adobe Research for helping set up the Adobe Firefly GenFill API. Boxiao Pan and Leonidas J. Guibas are supported by a grant from the Stanford Human-Centered AI Institute (HAI) and a Vannevar Bush Faculty Fellowship.

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

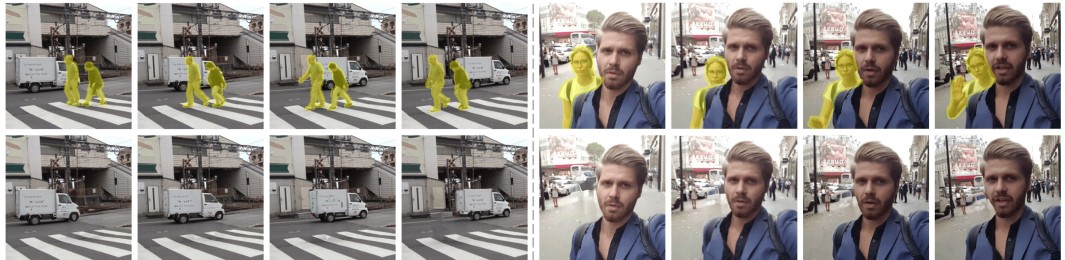

Figure 5: **Zero-shot video inpainting.** We show two cases from DAVIS, each with four sampled frames. The yellow regions denote the masked areas to be inpainted.

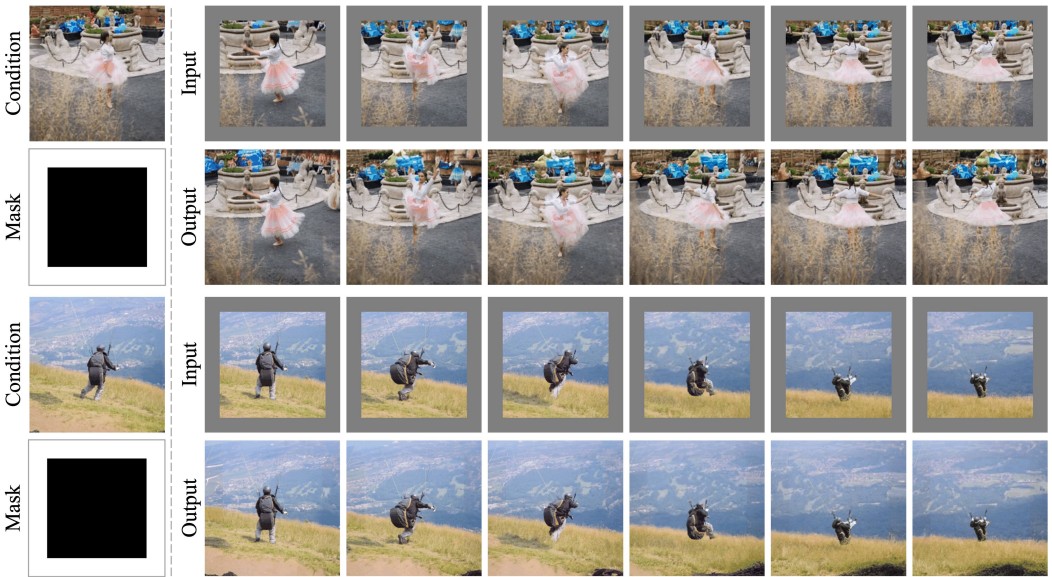

Figure 6: **Zero-shot video outpainting.** Both examples are from DAVIS.

# 7 Appendix

In this appendix, we first provide additional results and analysis in section 7.1, including inpainting / outpainting capability emerging from our model, generalization to domains different from the training data, robustness to inaccurate foreground segmentation, and inference runtime. We then describe essential processing steps for training and evaluation data in section 7.2. Next, we show failure cases and discuss limitations of our model in section 7.3. Lastly, we conclude by discussing the ethical impact of this work in section 7.4.

## 7.1 Additional Results and Analysis

**General video inpainting / outpainting**. Interestingly, once trained, certain general video inpainting / outpainting capability emerges from our model. We perform preliminary experiments by manually creating a mask sequence, and pass those with the foreground sequence as the input to our model. Two cases are shown in fig. 5, where our model is able to inpaint the missing regions, despite not explicitly trained so. Similarly, our model can also be applied to general video outpainting, whose results are shown in fig. 6. Specifically, we resize the original sequence of frames by 0.75, and pad them with gray boundaries. Associated masks are also created to indicate the missing regions. We then randomly select one frame and use Adobe Firefly [2] to outpaint it, with which as condition we outpaint the entire sequence.

These results may suggest that our model learns to approximate the underlying data distribution to a certain degree, possibly benefiting from the random condition dropping during training (section 3.4).

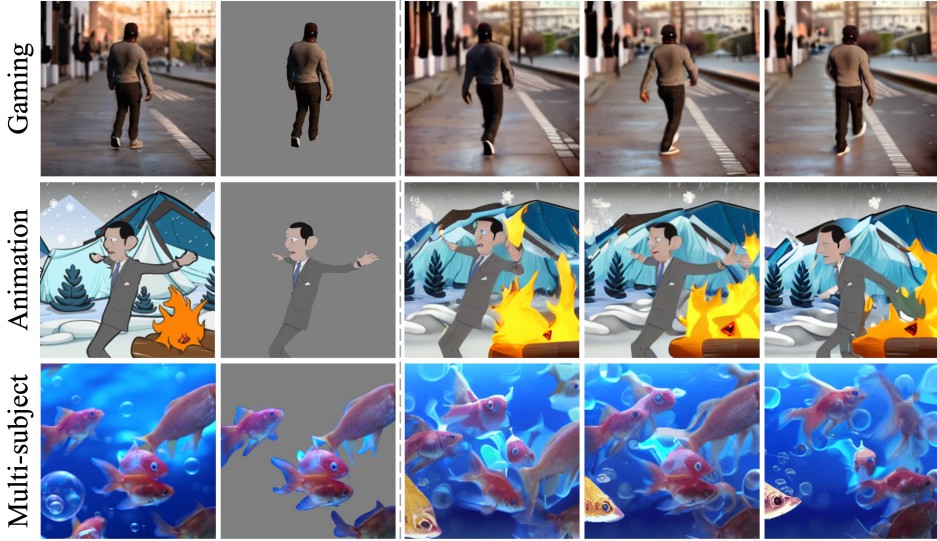

Figure 7: **Videos from various domains**. Gaming video from GTA.

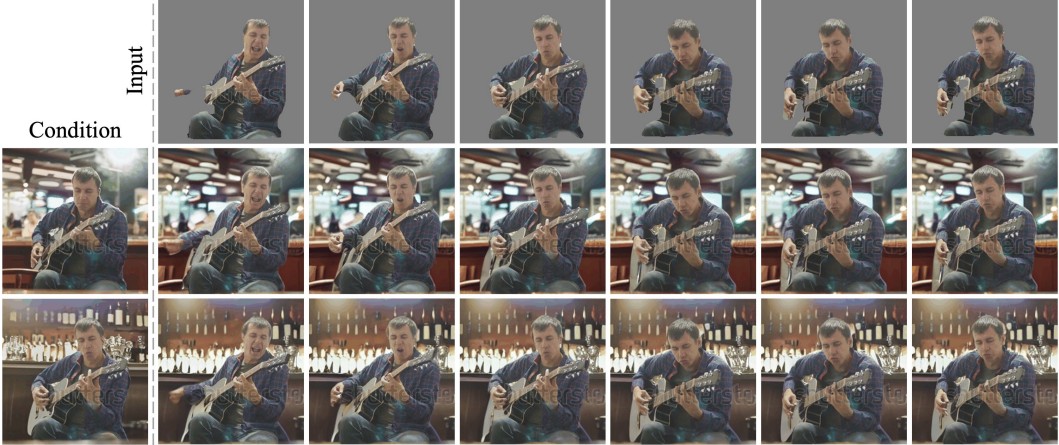

Figure 8: Our model is robust to inaccurate masks. We show one video sequence from HiC+ with two different condition frames, followed by six generated frames for each.

**Videos from various domains**. Our model generalizes to various domains such as gaming and animation clips, as well as videos with multiple moving subjects. In fig. 7, we show the condition frame with its segmentation, along with three generated frames for one example in each domain.

**Robust to inaccurate masks**. As stated in section 3.4, masks created or extracted in practice are often imperfect, being either incomplete or excessive. Here we show that our model trained in our designed procedure is robust to imperfect masks. In fig. 8 we showcase an example of this. Despite a large region of the guitar missing, our model is able to hallucinate them in a reasonable way by taking the global context into account.

**Runtime**. Generating one video on an NVIDIA A100 GPU takes about 8.5 seconds, enabling much faster idea iteration compared to traditional workflows.

## 7.2 Data Processing

**Training**. In Sec. 3.4 of the main manuscript, we described our data processing and augmentation strategies for training data. Specifically, to deal with incomplete segmentation, we apply random rectangular cut-outs to the segmentation and masks. We show two examples in fig. 9.

**Evaluation**. As mentioned in Sec. 1 of the main manuscript, at test time, the composited foreground-background frames used as condition can be created with various methods, such as photo editing

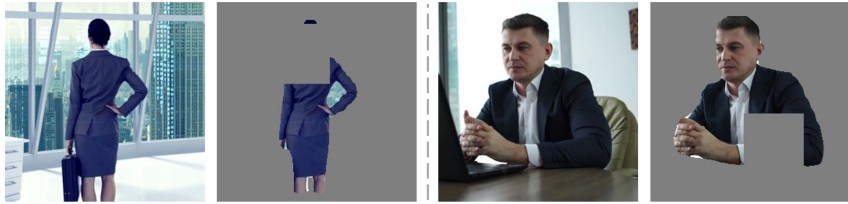

Figure 9: **Data augmentation**. We apply random cut-outs to the person segmentation during training. Here we show two examples of cut-outs with their corresponding original frames.

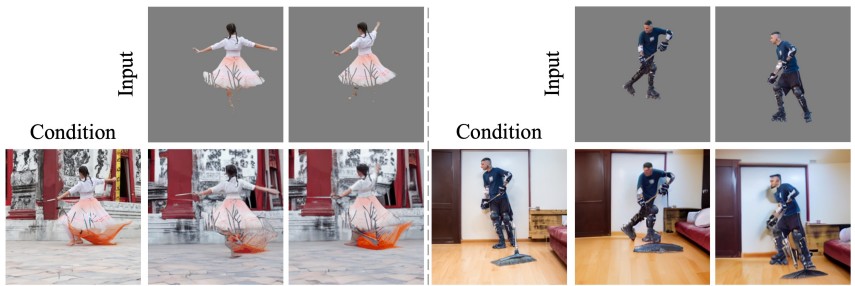

Figure 10: **Failure cases**. Foreground sequences from DAVIS.

tools (*e.g.* Adobe Photoshop [3]) or automated image outpainting methods (*e.g.* Dall-E [32]). In our experiments, we adopt ChatGPT 4 [29] and Adobe Firefly [2] to automatically synthesize these composited frames for use at test time. Specifically, we first sample the 0-th, 8-th, and 15-th frames from the input video, and ask ChatGPT the following question "Based on these three images, can you give me a description as prompt, less than 10 words, about one contextual scenario we can put this human in?". We then use Firefly to synthesize the outpainted frames, given the foreground segmentation and the text prompt. We use the "gpt4-vision-preview" version of ChatGPT 4 [‖], and the official Firefly GenFill [**].

### 7.3 Limitations

We show two failure cases of our method in fig. 10. In the first example, the grass-like texture on the dress is excluded from the segmentation, hence the model mistakenly perceives it to be an independent object growing outside the dress. While in the second example, the Firefly-inpainted frame has the broomstick facing towards the wrong direction. Although the model tries to generate something reasonable, it fails to correct this mistake to produce a coherent video. Despite certain fault tolerance from our model, providing proper input segmentation and condition signal helps ensure high-quality generation results.

### 7.4 Data Ethics

The HiC+ dataset [26] includes public datasets HVU [12], Charades [37], Moments [28], and WebVid10M [4]. Readers are encouraged to refer to Section A.1 in the supplementary material of [7] for the license of these datasets. Moreover, same as [26], more internal videos are incorporated during training and evaluation. We have conducted the necessary legal review process, and can provide more details of this data upon request.

We present a method that can synthesize highly realistic videos that place human and non-human subjects into diverse background scenes. While our work has implications and values for a wide range of fields, we note that our model can be misused to generate malicious content. Similar to Firefly GenFill [2] and Gen1 [13], the generated content can be watermarked to prevent misuse. Moreover, our model inherits the demographic biases presented in the training data. We make our best effort to demonstrate impartiality in all the results shown.

---

[‖]https://platform.openai.com/docs/overview
[**]https://firefly.adobe.com/generate/inpaint

