# OpenReview forum: "ActAnywhere: Subject-Aware Video Background Generation"
_NeurIPS.cc/2024/Conference — NeurIPS 2024 poster_

### Official Review · Reviewer_ghM9 · 2024-07-01

**Soundness:** 3
**Presentation:** 4
**Contribution:** 2
**Rating:** 5
**Confidence:** 5

**Summary:**

The task addressed by this paper is, given the the appearance and segmentation mask of a foreground subject in a video, for example a human running, to synthesize video backgrounds that are plausible and realistic, both in content and motion. For example, for a person running, if the ground is wet there should be splashes corresponding to footfalls, and the background scene should move along with the runner's movement.

The authors address this problem as a generative video outpainting task. They model it using a latent video diffusion model. The foreground subject appearances (encoded) and masks are concatenated with the latent feature vector, while the conditioning image (what the background should look like) is encoded via CLIP and is provided via cross-attention to the U-net.

A variety of qualitative results are included, demonstrating the ability to out-paint plausible scenes, including background motion, effects (splashing water) and even objects that the subject interacts with. Quantitative evaluations include ablations validating the main components of the model, as well as human rater studies comparing the quality of the output to competing methods.

**Strengths:**

Originality & Significance.
The paper addresses a minor new variant on the outpainting/inpainting problem, which introduces some new challenges such as picking up on foreground contextual cues and using that to inform the background.

Quality & Clarity:
The paper is clearly written, seems correct, and demonstrates visually appealing results with a variety of different foreground subjects (person, car, duck, etc.). The supplemental videos help illustrate the model's capabilities well. The empirical results also support the quality of the generative model.

**Weaknesses:**

The main weakness of this paper is very limited novelty, which compromises the claimed contributions of the paper.

Video inpainting/outpainting is a fairly well-known challenge. The main novelty here is that the masks are simply inverted: instead of deleting an unwanted person and generating the missing background pixels, here we retain only the desired foreground person and generate the missing pixels. I am not convinced that this difference is substantively novel.

Similarly, the modeling approach bears considerable similarity to [13] "Structure and Content-Guided Video Synthesis with Diffusion Models". In particular, compare Fig.2 from the two papers. The primary difference between the two is that this work conditions on foreground appearances and masks, while [13] conditions on estimated depth images. Again, the novelty is minor and unsurprising.

**Questions:**

Have you tried unconditional generation, i.e. not providing any signal as to what the desired background should be? This could be quite interesting, since it would challenge the model to understand exactly what the foreground subject is doing and to invent plausible backgrounds. For example, this child seems to be interacting with an object, what could that object be? This woman is riding something… what fits?


Would you like to discuss the appearance of the Shutterstock watermarks in the generated backgrounds, particularly in figure 3? This could be an interesting example of the model using foreground context informing the background generation, as per my previous question.

**Limitations:**

Yes.

---

> ### Author Rebuttal · Authors · 2024-08-07
>
> We thank the reviewer for their thoughtful feedback and for acknowledging that the introduced problem possesses originality, the high quality of the results demonstrates the effectiveness of the proposed model, and that the paper is clearly written. We next address the reviewer’s questions and comments.
>
> > Video inpainting / outpainting is a fairly well-known challenge…
>
> As discussed in L40-L44 and L100-L112 of the submission, the key challenge of our task compared to general video inpainting / outpainting lies in **generating dynamic interactions** and **large background regions** that **follow the image condition**. This is acknowledged by reviewers Pq5G and gAcX. It is nontrivial how to extend the video inpainting / outpainting frameworks, which generally tackle a pixel harmonization problem, to solve these challenges. Also, in Appendix 6.1, we showed that our model, once trained, exhibits general video inpainting / outpainting capabilities, while general inpainting / outpainting methods are not able to solve our proposed task.
>
> > The modeling approach is very similar to Gen1 [13]
>
> Our key contributions include 1) introducing the novel problem of automated subject-aware video background generation with a background image as condition, and 2) proposing a specific video diffusion-based pipeline to solve this problem. Prior works such as Gen1 [13] are not able to solve the introduced problem, as demonstrated qualitatively and quantitatively in Sections 4.2 and 4.3 of the main manuscript. We found empirically that our proposed framework can effectively tackle the problem.
>
> > Unconditional generation with the model
>
> Thank you for the question. We have performed the requested experiment and included the results in Fig. 2 of the attached PDF. Specifically, we set the condition to zeros, same as the random condition dropping described in L192-L194 of the submission. We sampled two subjects from the HiC+ dataset, and for each we ran our model with three different seeds. From each of the three generated videos, we selected one frame to show its input segmentation along with the generation. We observed that the model can generate reasonable random backgrounds that fit the foreground.
>
> > Watermark in the generated backgrounds
>
> Webvid* videos are a major source of the HiC+ dataset, which all contain the “shutterstock” watermark. Training on them provides a dataset bias such that when conditioned on segmentations or a condition frame with the watermark, the generated results will also contain and complete (if the watermark appears partially in the input) the watermark. As the reviewer pointed out, this is an example of the model using foreground context to inform the background generation.
>
> Despite that this issue is orthogonal to the focus of this work, we are aware of methods that can alleviate watermarks appearance in the generation, e.g. the “Domain Adapter” in AnimateDiff [17]. We leave the integration of such methods in our framework to future work.
>
> *Frozen in Time: A Joint Video and Image Encoder for End to End Paper. Bain et al. ICCV 2021.

---

> > ### Comment · Reviewer_ghM9 · 2024-08-09
> >
> > Thank you for exploring unconditional background generation. I think this is an intriguing variant of the problem to showcase. I like how the model attempts to correct for the overly-tight cropping of the foreground person, by generating the missing shoe.
> >
> > Given the general agreement from other reviewers that this problem is novel and interesting, I'll retract that part of my assessment, and boost the score.

---

> > > ### Author Response · Authors · 2024-08-09
> > >
> > > We are glad to hear that we addressed your concerns. We will include the unconditional generation results in the final version of the paper. Thank you for raising the score! And thanks again for your efforts spent reviewing our paper!

---

### Official Review · Reviewer_gAcX · 2024-07-11

**Soundness:** 4
**Presentation:** 4
**Contribution:** 3
**Rating:** 7
**Confidence:** 4

**Summary:**

This paper introduces ActAnywhere, a video diffusion model designed to generate video backgrounds that adapt to the foreground subject's motion. By utilizing a sequence of foreground subject segmentation and a background image, the model produces realistic videos with coherent foreground-background interactions. Experiments on a large-scale dataset demonstrate the model's effectiveness, outperforming existing methods in generating realistic and dynamic backgrounds.

**Strengths:**

S1: The paper introduces a novel problem of automated subject-aware video background generation.

S2: The methodology shows improvements in generating coherent videos with realistic subject-background interactions.

S4: The contributions are significant, particularly for applications in the movie industry and visual effects.

S4: The paper is comprehensive and well-written.

**Weaknesses:**

W1: The paper lacks sufficient comparison with a broader range of existing methods, particularly those leveraging recent advancements in video generation and editing (though they are not for background generation, they can do).

W2: It seems that the model relies heavily on the quality of the foreground video segmentation masks.

**Questions:**

Q1: I wonder about the impact of the quality of the foreground segmentation. It seems that the model relies heavily on the quality of the foreground segmentation.

Q2:  How scalable is your model for generating longer video sequences? Have you tested its performance in generating videos of varying lengths, and what are the results?

**Limitations:**

I think there is no potential negative societal impact and the authors have addressed the limitations.

---

> ### Author Rebuttal · Authors · 2024-08-07
>
> We appreciate the positive feedback from the reviewer and thank them for acknowledging that our introduced problem is novel, the proposed method is effective and makes a significant contribution to the movie and VFX industries, and that our paper is well-written. We address the reviewer's individual questions and comments below.
>
> > Lacks sufficient comparison with recent works on video generation and editing
>
> To the best of our knowledge, the work with the closest setting to ours is AVID*, which can perform a task termed “Environment Swap” by editing the background region with a text condition. However, AVID does not take image conditioning, and hence cannot make the generated video follow the strict guidance specified by an image as we do. Moreover, from the results the authors showed on the website, they do not manifest realistic foreground-background interactions as ours (e.g. the sand does not deform according to the woman / tiger’s movement), and fail to generate correct shadows (the shadows are simply carried over from the input). We also note that this work was published recently in CVPR which was after the NeurIPS submission deadline, and is not open sourced yet which makes it hard to compare with.
>
> Apart from this work, we believe others study different settings from ours and are generally non-trivial to extend to work under our setting. We are happy to compare any specific methods that the reviewer may suggest.
>
> *AVID: Any-Length Video Inpainting with Diffusion Model. Zhang et al. CVPR 2024.
>
>
> > It seems that the model relies heavily on the quality of the foreground segmentation
>
> As referenced in L204 of the main manuscript and discussed in Appendix 6.1, our model is in fact quite robust to inaccurate masks (Fig. 8), thanks to our designed data augmentation and processing procedures (i.e. random cut-outs and image erosion to the segmentation and mask) as noted in Sec. 3.4 of the main manuscript and Appendix 6.2.
>
> > How scalable is the model for generating longer / variable-length video sequences?
>
> Due to the modular nature of our framework, we can easily scale up to generating longer videos if we swap to a different base model. The choice of generating 16-frame videos aligns with many previous works (e.g. [9, 17] for 16-frame generations, and [24] for 8-frame generations) and is primarily limited by the compute resource we had. Generating longer videos is out of the scope of this work, and we will explore this in future work with the latest DiT backbone.

---

> > ### Comment · Area_Chair_Z54u · 2024-08-12
> > **Has the rebuttal addressed your concerns?**
> >
> > Dear Reviewer gAcX,
> >
> > Thank you again for your time to review this paper. Could you please check if the authors' rebuttal has addressed your concerns at your earliest convenience? The deadline of the discussion period will end in about 24 hours. Thank you!
> >
> > Best regards,
> >
> > AC

---

### Official Review · Reviewer_Pq5G · 2024-07-13

**Soundness:** 3
**Presentation:** 2
**Contribution:** 3
**Rating:** 5
**Confidence:** 4

**Summary:**

This paper studies a new topic: automatic background generation of moving foreground subject. Different from video inpainting/outpainting and other video editing methods, the method in this paper can maintain the consistency of foreground moving subject, and maintain reasonable and realistic interactions, camera motion, lighting and shadows.

It can also generalizes to diverse scenarios including non-human objects, gaming and animations, as well as videos with multiple moving subjects. Input a foreground segmentation image sequence and a foreground mask sequence, and under the guidance of a background reference image, generate a motion video of the interaction between the foreground and the background. Specifically, the spatial layer parameters of Denoising 3D Unet are loaded from a pre-trained SD-inpainting model, and the temporal layer(motion module) is loaded from the pre trained model of AnimateDiff V2. The input is a feature map of nine channels, which concatenates the nosie latent, the vae encoder feature of the foreground segmentation map, and the downsampled foreground mask in the channel dimension.

The CLIP feature of the background reference map is injected into the Unet through cross attention.Trained on the HiC+ dataset (2.4M videos of human-scene interactions), Given the error of foreground segmentation, random rectangular cut-outs augmentation is applied to the foreground segmentation sequence and foreground mask sequence

**Strengths:**

A new task was studied: automatic background generation of moving foreground, and reasonable and realistic foreground/background interaction, camera motion, lighting and shadows were required.

Although it is a new task, the results of similar methods were compared under fair conditions as much as possible, which indeed proved the superiority of the method in this paper.

**Weaknesses:**

The training resolution is low (256x256), and it is unclear whether the method in this paper still has advantages and reliability under high-score conditions?
Ablation experiment: The method of using background reference image for guidance does not seem to be fully considered, such as the method similar to IPAdapter?

The training details are not clear: is it fine-tuning the spatial layer and the motion module at the same time? Or only fine-tuning the motion module? Or is cotraining still required?

This article spends a certain amount of space to explain the difference in results between the method proposed in this paper and the SD-inpainting method, but in terms of the algorithm pipeline, except for using the CLIP features of the background reference image for guidance, the rest is almost exactly the same as the SD-inpainting model, and even the spatial layer of Unet directly loads the pre-trained model of SD-inpainting.

The article does not seem to explain the difference in principle between the method proposed in this paper and the SD-inpainting method?

**Questions:**

Please refere weakness

**Limitations:**

Yes.

---

> ### Author Rebuttal · Authors · 2024-08-07
>
> We thank the reviewer for their positive feedback and for acknowledging that our introduced task is novel, and that the experimental comparisons are under a fair setup. We address the reviewer’s particular questions and comments below.
>
> > Training resolution is low… The method of using reference image does not seem fully considered, e.g. the method similar to IP-Adapter?
>
> There is no fundamental limitation for our method to work under higher-resolution settings, granted a base model under a higher resolution and more compute resources.
>
> Compared to the method of condition encoding in IP-Adapter, we only support image conditioning and thus do not need to map image and text to a joint space, hence we do not need the Decoupled Cross-Attention module as in their framework. Similar to the image encoding method in IP-Adapter though, we also have a linear layer to project the CLIP image features to a lower dimensional feature space, as noted in L179-L182 of the main manuscript.
>
> > Are we finetuning only the spatial layers, only the motion module, or both?
>
> Thank you for the question. We finetune both the spatial layers and the motion module layers at the same time. We will clarify this in the final version of the paper.
>
> > Technical framework is very similar to SD-inpainting
>
> Our key contributions include 1) the introduction of the novel problem of automated subject-aware video background generation with a background image as condition, and 2) the proposal of a specific video diffusion-based pipeline that is carefully designed and tailored to solve this problem.
> SD-inpainting works only under the image setting, and it is unclear how to extend it to videos. Our proposed framework, along with the designed self-supervised training pipeline and data augmentation strategies, contribute to our solution altogether.

---

> > ### Comment · Area_Chair_Z54u · 2024-08-12
> > **Has the rebuttal addressed your concerns?**
> >
> > Dear Reviewer Pq5G,
> >
> > Thank you again for your time to review this paper. Could you please check if the authors' rebuttal has addressed your concerns at your earliest convenience? The deadline of the discussion period will end in about 24 hours. Thank you!
> >
> > Best regards,
> >
> > AC

---

> > > ### Comment · Reviewer_Pq5G · 2024-08-12
> > >
> > > Thanks authors for providing answers. Most of my concerns was addressed. But I still think the technical contribution is not very significant. I kept my score as 5.

---

> > > > ### Author Response · Authors · 2024-08-13
> > > >
> > > > We are glad to hear that most of your concerns have been addressed. We would like to note again that we introduced a novel problem with practical implications (as acknowledged by Reviewer gAcX) that no existing methods can tackle to a reasonable extent. We also demonstrated that our proposed framework surpasses all compared baselines by significant margins.
> > > >
> > > > Thank you again for your efforts spent reviewing our paper!

---

### Official Review · Reviewer_Aquh · 2024-07-15

**Soundness:** 4
**Presentation:** 3
**Contribution:** 2
**Rating:** 5
**Confidence:** 5

**Summary:**

This paper study to automatically generate video background that tailors to foreground subject motion. It proposes ActAnywhere, a video diffusion model that takes as input a sequence of foreground subject segmentation and an image of a novel background and generates a video of the subject interacting in this background. Both quantitative and qualitative comparisons demonstrate that the model significantly outperforms existing methods, which fail to accomplish the studied task.

**Strengths:**

1. The task is interesting and the result seems to be very competitive.

2. The paper is written clear.

**Weaknesses:**

1. Only encoder condition frame by a clip encoder cannot contains details information, especially for high-resolution input images. Please refer AnimtaeAnyone for human image animation. And please compare such a referencenet attention way.

2. Can you provide some video demos. I suppose there may exists blur boundary results.

3. Since the task seems to be very related to human animation task, could you please show some results on TikToK dataset? Please see AnimateAnyone paper for comparision details.

**Questions:**

See weakness

**Limitations:**

See weakness

---

> ### Author Rebuttal · Authors · 2024-08-07
>
> We appreciate the positive feedback from the reviewer and thank them for acknowledging that our introduced task is interesting and that our proposed model achieved competitive results. We next address the reviewer's individual questions and comments.
>
> > CLIP encoder alone cannot contain detailed information… Compare to the approach of AnimateAnyone using ReferenceNet.
>
> We thank the reviewer for the suggestion. We would like to note that detail preservation of the condition is still an open question. And because our framework is fairly general and agnostic to the image encoder, we can swap the CLIP encoder with any encoder with ease, e.g., DINO* if object-centric features are more desirable. ReferenceNet is an interesting alternative, but AnimateAnyone is not open sourced, which makes it hard to directly compare. Nonetheless, we are happy to compare if the reviewer insists.
>
> *Emerging Properties in Self-Supervised Vision Transformers. Caron et al. ICCV 2021.
>
> *DINOv2: Learning Robust Visual Features without Supervision. Oquab et al. TMLR, 2024.
>
> > Provide video demos. Blurry boundaries may exist in the videos.
>
> We already included extensive video results in the easily accessible supplementary webpage, as mentioned in L206-L208 of the main manuscript. We did not observe any particular cases with blurry boundaries.
>
> > Task relevant to human animation. Show results on the TikTok dataset.
>
> We note a key difference from the human animation task that the goal is in fact the opposite - we are given the foreground motion and try to generate the moving background. The TikTok dataset has a different focus to generate the foreground motion instead. Hence, the background in the dataset is often very simple and does not move much, and very little, if any, interaction happens between the foreground and the background. Thus the dataset is not a good fit for our task. And since AnimateAnyone is not open sourced, it is hard to compare experimentally. Nonetheless, as requested by the reviewer, we tested our model on two samples from the TikTok dataset and included the results in Fig. 1 of the attached PDF. Our model demonstrates good generalization over the tested data.

---

> > ### Comment · Area_Chair_Z54u · 2024-08-12
> > **Has the rebuttal addressed your concerns?**
> >
> > Dear Reviewer Aquh,
> >
> > Thank you again for your time to review this paper. Could you please check if the authors' rebuttal has addressed your concerns at your earliest convenience? The deadline of the discussion period will end in about 24 hours. Thank you!
> >
> > Best regards,
> >
> > AC

---

### Author Rebuttal · Authors · 2024-08-07

We would like to sincerely thank the AC and the reviewers for their hard work and time in reviewing our submission. We appreciate the positive feedback and recognition of the novelty and significance of the introduced problem, the high quality of the results and the effectiveness of the proposed method, as well as the clear paper writing. We also appreciate the insightful suggestions for further improving our work.

We also want to emphasize that the key technical contributions of this paper are 1) the introduction of the novel problem of automated subject-aware video background generation, and 2) the proposal of a video diffusion-based pipeline that is carefully designed and tailored to solve this problem. And as we have shown in Sec. 4 of the main manuscript, no baseline was able to effectively solve the introduced problem (and with reasonable adaptations), and our proposed method achieved significantly better performance than the baselines both qualitatively and quantitatively. This demonstrates the challenge of the introduced problem and the effectiveness of the proposed solution.

In the comments below, we have addressed all specific issues / concerns / questions raised by the four reviewers by replying to each individually. We also attached a PDF which contains the additional results requested by Reviewers **Aquh** and **ghM9**.

Thank you again for your time and efforts throughout the review process -- they are greatly appreciated!

---

### Comment · Area_Chair_Z54u · 2024-08-09

Dear Reviewers,

Thank you very much again for your valuable service to the NeurIPS community.

As the authors have provided detailed responses, it would be great if you could check them and see if your concerns have been addressed. Your prompt feedback would provide an opportunity for the authors to offer additional clarifications if needed.

Best regards,

AC

---

### Decision · Program_Chairs · 2024-09-25

**Decision:**

Accept (poster)

**Comment:**

The significance of this paper is unanimously recognized by all the reviewers, including the novelty of the proposed task, its impact in other applications (e.g., movie industry), thorough experimental evaluation and comparisons with other methods, and clear writing of the paper. The AC thus recommends to accept this paper as a Poster.